# Unusual Presentation in WAGR Syndrome: Expanding the Phenotypic and Genotypic Spectrum of the Diseases

**DOI:** 10.3390/genes13081431

**Published:** 2022-08-12

**Authors:** Qiwei Wang, Xulin Zhang, Tingfeng Qin, Dongni Wang, Xiaoshan Lin, Yuanyuan Zhu, Haowen Tan, Lanqin Zhao, Jing Li, Zhuoling Lin, Haotian Lin, Weirong Chen

**Affiliations:** 1State Key Laboratory of Ophthalmology, Guangdong Provincial Key Laboratory of Ophthalmology and Visual Science, Guangdong Provincial Clinical Research Centre for Ocular Diseases, Zhongshan Ophthalmic Centre, Sun Yat-sen University, Guangzhou 510060, China; 2Aegicare, 3803 Building 11A, Shenzhen Bay Ecological Technology Park, Nanshan District, Shenzhen 518063, China

**Keywords:** WAGR syndrome, copy number variation, severe lens abnormality, *ARL14EP*

## Abstract

The deletion of chromosome 11p13 involving the *WT1* and *PAX6* genes has been shown to cause WAGR syndrome (OMIM #194072), a rare genetic disorder that features Wilms’ tumor, aniridia, genitourinary anomalies, as well as mental retardation. In this study, we expand the genotypic and phenotypic spectrum of WAGR syndrome by reporting on six patients from six unrelated families with different de novo deletions located on chromosome 11p13. Very rare phenotypes of lens automated absorption and lens thinning were detected in four of the six patients. We assessed the involvement of the *ARL14EP* gene in patients with and without severe lens abnormalities and found that its deletion may worsen the lens abnormalities in these patients.

## 1. Introduction

WAGR syndrome (OMIM #194072) is a rare genetic disorder with a birth prevalence of 1/500,000 [1]. Its hallmark characteristics include an increased risk for Wilms tumor, total or partial aniridia, genitourinary anomalies (ranging from sexual ambiguity to ectopia testis), and mental retardation (variable degrees of intellectual disability). Other findings may include obesity (WAGRO, OMIM #612469), glaucoma, cataracts, kidney failure, congenital diaphragmatic hernia, and autism [2,3,4]. The inheritance pattern of WAGR syndrome is autosomal dominant; moreover, the preferentially paternal origin of the 11p13 chromosome deletion is observed in the families’ genotypes [5].

The deletion of chromosome 11p13 involving the *WT1* and the *PAX6* genes has been shown to cause WAGR syndrome [6]. The increased risks for Wilms tumor and genitourinary anomalies in WAGR patients are attributed to the deletion of *WT1*. Aniridia and associated ocular defects are triggered by a *PAX6* haploinsufficiency that encodes a transcriptional regulator required for the eye development [6]. Furthermore, brain-derived neurotrophic factor (*BDNF*, OMIM #113505) has been proven to be associated with intellectual disability, autism, and childhood-onset obesity in subjects with WAGR [4,7]. Theoretically, the severity of the phenotype depends on the size of the deletion. A deletion map of the WAGR region, ranging from 11p14 to 11p11 [1,8], can be found in earlier studies [1,9]. However, the relationship between the clinical phenotypes and genotypes of WAGR is not fully understood. Patients with a similar deletion region may present with different phenotypes and vice versa [9].

Herein, we present the cases of six patients affected by typical and atypical WAGR syndrome from six unrelated families, which reveal rare phenotypes of keratolenticular adhesion, lens thinning, and partial lens material absorption. To the best of our knowledge, there is no previous report of an association between abnormal lens thinning and partial lens material absorption with WAGR syndrome. Our findings demonstrate a new phenotype, thus expanding the genotype spectrum of this severe disorder.

## 2. Materials and Methods

### 2.1. Patients

This study is part of a series of ongoing studies at the Childhood Cataract Program of the Chinese Ministry of Health (CCPMOH) [10]. The inclusion criteria were a confirmation of congenital cataracts (CCs) and the completion of whole exome sequencing (WES). A total of 269 CCs patients from the Home for Cataract Children of the Zhongshan Ophthalmic Center seen between March 2016 and January 2021, and their available family members, were included. The study was approved by the Institutional Review Board of the Zhongshan Ophthalmic Center adhering to the tenets of the Declaration of Helsinki. Written informed consent was obtained from the legal guardian of each child.

### 2.2. Clinical Assessments

Clinical data were collected based on a thorough ocular examination and medical history review. Facial photographs were taken with a digital camera. Anterior segment photographs were taken using slit-lamp photography (BX900; HAAG-STREIT AG, Bern, Switzerland). Ultrasound biomicroscopy (UBM) (SW-3200L; Tianjin Suowei Electronic Technology Co., Ltd., Tianjin, China), fundus photography (VISUCAM 224, Carl Zeiss Meditec AG, Jena, Germany), B-scan ultrasound (Aviso, Quantel Médical, Clermont-Ferrand, France), and optical coherence tomography (OCT) (CIRRUS HD-OCT 5000, Carl Zeiss Meditec AG, Jena, Germany) were performed to screen the anterior and posterior segment disorders. Detailed clinical data, including basic information, family history, and other medical history, were recorded in case report forms.

### 2.3. Exome Sequencing and Bioinformatic Analysis

Genomic DNA was extracted from peripheral blood samples of patients using QIAamp DNA Mini Kit (Qiagen, Hilden, Germany). WES was performed in all of the probands using the Agilent v6 targeted sequence capture library process method with the Hiseq 2500 platform (Illumina Inc., San Diego, CA, USA) as described previously [11]. A copy number variation (CNV) kit was applied to our data [12]. During CNV calling, two event-segmentation algorithms (circular binary segmentation and the hidden Markov model for germline (hmm-germline)) were chosen for better recognition of both larger and smaller CNV events. Two types of segmentation outcomes were merged and labeled for subsequent analysis. An in-house tool utilizing both copy numbers and B-allele frequencies aligned against chromosome coordinates was designed for CNV event visualization. The threshold parameter for copy number reversion from the log2 ratio was altered to “−1.6, −0.8, 0.5, 1,” which was tested, giving better results than the default value. The B-allele frequencies were computed using the samtools mpileup tool. The AnnotSV program and corresponding annotation databases were locally installed to annotate detected CNV events on each sample for clinical interpretation [13].

In CNV calling, a stable and reliable CNV reference is critical. Here, an iterative approach was used to train the reference for CNV identification. First, a set of 60 samples over the same sequencing protocol was enrolled to create an initial reference. Then, CNV calling was run over this reference for each training sample. Any samples with any events over 1 Mbp were excluded from the training set. New samples were added to retain 60 samples in the training set for each iteration. The iteration ended once more than 30 samples were left after exclusion. Then, the final reference was built on that sample set.

### 2.4. Statistical Methods 

The probands with a thinning lens or an absorbed cloudy lens were allocated into the severe lens abnormality group. To analyze the genotype–phenotype correlations of the putative effects of the candidate deleted genes on the risk for severe lens abnormality, a two-sided Fisher’s exact test was performed for categorical variables using IBM SPSS Statistics 25 (IBM Corporation, Armonk, NY, USA). A probability value of less than 0.05 indicated a rejection of the null hypothesis, such that the severe lens abnormality was significantly more frequently observed in probands with a particular gene deletion [1].

## 3. Results

In total, six different-sized de novo deletions located at 11p13 from six probands were identified from our exon sequencing data of 269 probands (Figure 1). All six patients showed aniridia and cataracts. Four of the six patients presented with Wilms tumor before the age of 2 years. Detailed ocular manifestations of the patients were collected (Table 1). Then, we attempted to build the phenotype–genotype correlations to explore the possible functions of the affected candidate genes.

### 3.1. Case 1

An 11-month-old girl presented with whitish pupils, and nystagmus was noticed by her parents at birth. A further examination found a thin lens with nuclear opacity of the right eye, normal thickness, posterior cortical cataract of the left eye (Figure 2A and Figure 3A), a cup-to-disc (C/D) ratio of 0.6, aniridia, and macular hypoplasia in both eyes (Figure 4A,B). The intraocular pressure (IOP) was 15.3 mmHg in the right eye and 16.2 mmHg in the left eye, as measured using a Tono-Pen (Reichert, Inc., Depew, NY, USA). The axial lens (AL) was 22.24 mm in the right eye and 22.98 mm in the left eye. She underwent uneventful bilateral nephron-sparing surgery due to Wilms tumor. Developmental delay was also diagnosed. CNV analysis based on WES found a deletion of Chr11:19372477–35827981. WAGR syndrome was confirmed. Ocular surgery was not prescribed. Twice-a-year follow-up was scheduled. At the 1-year follow-up, the middle of the nuclear opacity of the right eye had been spontaneously absorbed (Figure 2B). The IOPs of both eyes were normal, and the tumor did not recur or metastasize.

### 3.2. Case 2

A 23-month-old girl presented with bilateral whitish cornea, and nystagmus was noticed by her parents at birth. A slit-lamp examination showed keratolenticular adhesion (the anterior surface of the thin lens adherent to the posterior surface of the cornea), corneal neovascularization, aniridia, and a thin lens with nuclear and bubble-like opacities in both eyes (Figure 2C and Figure 3B). An OCT examination revealed macular hypoplasia in both eyes. The AL was 16.63 and 17.07 mm in the right and left eyes, respectively, indicating nanophthalmos. The IOP was 16 mmHg in the right eye and 35 mmHg in the left eye, as measured using a Tono-Pen. The IOP was controlled using ocular IOP-lowering medications. The girl underwent uneventful right kidney resection due to Wilms tumor. An atrial septal defect and developmental delay were also diagnosed. A CNV analysis based on WES found the deletion of Chr11:26210680–35006316. WAGR syndrome was confirmed. Ocular surgery was not prescribed. Twice-a-year follow-up was scheduled. At the 1-year follow-up, the ocular disease had not deteriorated further, and the tumor did not recur or metastasize.

### 3.3. Case 3

An 18-month-old boy presented with bilateral whitish cornea, and nystagmus was noticed by his parents at birth. Slit-lamp examination showed keratolenticular adhesion, corneal neovascularization, aniridia, and a thin lens with nuclear and bubble-like opacities. The anterior surface of the lens was adherent to the posterior surface of the cornea (Figure 2D and Figure 3C). An OCT examination indicated macular hypoplasia. The IOP was 18 mmHg in the right eye and 17 mmHg in the left eye, as measured using a Tono-Pen. Wilms tumor and developmental delay were diagnosed when the boy was 15 months old. He underwent uneventful left kidney resection and postoperative radiotherapy. A CNV analysis based on WES found a Chr11:30032154–36680772 deletion. WAGR syndrome was confirmed. Ocular surgery was not prescribed. Twice-a-year follow-up was scheduled. During the 2-year follow-up, the IOP of the child was normal, the ocular disease did not deteriorate, and the tumor did not recur or metastasize.

### 3.4. Case 4

A 16-month-old girl presented with a whitish cornea, which was noticed by her parents at birth. A further examination found keratolenticular adhesion, corneal neovascularization, aniridia, cortical cataract, and macular hypoplasia in both eyes (Figure 2E). Visual acuity UBM, OCT, fundus photography, and AL measurements were unavailable due to noncooperation. The IOP was 14.3 mmHg in the right eye and 12 mmHg in the left eye as measured using a Tono-Pen. Developmental delay was diagnosed. No kidney or genitourinary anomality was detected after a careful systemic examination. A CNV analysis based on WES found a deletion of Chr11:30358107–43941673. WAGR syndrome was confirmed. Ocular surgery was not prescribed. Twice-a-year follow-up was scheduled. At the 1-year follow-up, the IOP of the child was normal, the ocular disease had not deteriorated, and no kidney tumor was detected.

### 3.5. Case 5

A 3-month-old girl presented with whitish pupils, which were noticed by her parents at birth. A further examination revealed nuclear cataracts and aniridia in both eyes (Figure 2F). The IOP was 12.2 mmHg in the right eye and 14.6 mmHg in the left eye, as measured using a Tono-Pen. The AL was 15.83 mm in the right eye and 13.98 mm in the left eye, indicating nanophthalmos. She underwent uneventful right kidney resection due to Wilms tumor, followed by cataract extraction surgery. The first month after cataract surgery, an increased IOP was detected in both eyes (right eye, 21.6 mmHg; left eye, 25.8 mmHg). IOP-lowering medication was prescribed. However, the parents did not give the child the medicine as directed and did not show up for the follow-up visits as required. Six months later, the girl showed an enlarged C/D ratio (0.5) (Figure 4C). A secondary glaucoma was diagnosed. A CNV analysis based on WES found a Chr11:30431595–36680772 deletion, thus confirming WAGR syndrome.

### 3.6. Case 6

A 10-year-old boy presented with nystagmus and bilateral subcapsular cataract (Figure 2G). A further examination found aniridia and macular hypoplasia (Figure 4D,E). The best-corrected visual acuity was 0.1 and 0.15, IOP was 13.3 and 14.2 mmHg, AL was 15.55 and 15.42 mm in the right and left eyes, respectively. Except for hypospadias, no other abnormality was detected after careful systemic examinations. A CNV analysis based on WES found a deletion of Chr11:31784908–33773171. WAGR syndrome was confirmed. Ocular surgery was not prescribed. Twice-a-year follow-up was scheduled.

### 3.7. Statistical Analysis

The *ARL14EP* gene was deleted in all four probands (case 1, case 2, case 3, and case 4) with severe lens abnormalities and was intact in the two probands without severe lens abnormality. Therefore, the influence of the *ARL14EP* gene was tested. No statistically significant correlation was observed (*p* = 0.067). 

## 4. Discussion

We identified six novel heterozygous deletions of the chromosome 11p13 region in 6 of 269 independent probands with CCs and made a definitive diagnosis of WAGR syndrome. The six cases presented are of rare phenotypes, including thinning of the lens, partial lens absorption, and keratolenticular adhesion with corneal neovascularization. The genotype–phenotype correlations of these cases were analyzed.

WAGR syndrome is an ideal model for providing direct evidence of specific gene function in humans, as it is a disease with naturally occurring chromosomal deletion of different sizes. We identified different heterozygous deletions affecting the *PAX6* and *WT1* genes on chromosome 11p13 in all six probands with WAGR syndrome. All of the probands presented with aniridia and cataracts. Four of the six probands indicated Wilms tumor. Three of the six probands were associated with glaucoma before or after cataract surgery. Three of the six probands presented with thin lens and keratolenticular adhesion with corneal neovascularization. Interestingly, the cloudy lens of one proband was partially absorbed during the follow-up when the patient was 2 years old. The phenotypes of lens automated absorption and lens thinning were classified into severe lens abnormality groups. The *ARL14EP* gene was deleted in all four probands with severe lens abnormalities and was intact in both of the two probands without severe lens abnormality. We postulate that the damage of the *ARL14EP* gene leads to an increased risk for severe lens abnormalities in patients with 11p13 deletions. According to our observations, simultaneous loss of the *PAX6* and *ARL14EP* genes by a single deletion in patients with WAGR syndrome may effectively act as a double hit that worsens the lens damage caused by the deletion of *PAX6*. In the genotype–phenotype correlation analysis, only a boundary correlation between *ARL14EP* haploinsufficiency and the presence of severe lens abnormalities were observed (*p* = 0.067). A methodologically sound and sufficiently sized trial is needed to further analyze the correlation.

*ARL14EP* (OMIM: 612295; ADP-ribosylation factor-like GTPase 14 effector protein), located on chromosome 11p13, is a gene involved in WAGR syndrome. It encodes a small nuclear protein (35 kDa) that interacts with ADP-ribosylation factor-like 14 (ARL14). *ARL14EP* has proven to be critical for transcallosal connectivity [14]. The haploinsufficiency of *ARL14EP* has been linked to callosal hypoplasia. The animal model shows that *ARL14EP* knockdown leads to transcallosal projection disruption. It has also been suggested that *ARL14EP* interacts with histone H3 lysine 9 methyltransferase MET-2 (SETDB), which initiates and regulates the timing of heterochromatin formation [15]. However, no study has explored the association between *ARL14EP* and ocular abnormalities. 

The phenotype of aniridia in patients with WAGR syndrome is caused by the haploinsufficiency of *PAX6* (OMIM: 607108) [16]. *PAX6* cis-regulatory element (SIMO) from intron 9 of *ELP4*, located next to the *PAX6* gene, is critical for the continued tissue-specific expression of *PAX6*. The mutation or deletion of the SIMO element causes the defective maintenance of the expression of the *PAX6* gene in the lens and therefore results in aniridia, even when the *PAX6* gene is intact [17]. Therefore, the concurrent deletion of the SIMO element and *PAX6* gene in WAGR syndrome leads to a more severe ocular phenotype [18]. Among the six patients in our cohort, the SIMO element was affected in all of the cases, with the exception of case 6. The ocular phenotype of case 6 was the mildest among the phenotypes observed; this particular patient (male) presented with subcapsular cataracts, macular hypoplasia, and nystagmus. The genotype–ocular phenotype correlation in our cohort supports the importance of SIMO elements in ocular development.

The increased risk for Wilms tumor in patients with WAGR syndrome is caused by a haploinsufficiency of *WT1* (OMIM: 607102); it has been suggested that *LMO2* gene haploinsufficiency contributes to a further increase in this risk, based on a genotype–phenotype correlation analysis [1]. In our cohort, there were four patients affected by Wilms tumor with *LMO2* deletion, one patient with *LMO2* deletion and no Wilms tumor until she was 3 years old, and one patient without *LMO2* deletion and Wilms tumor until he was 11 years old. For the cases with *LMO2* deletion, a close surveillance of the kidneys is needed.

Despite the fact that adding a CNV analysis in WES may improve the overall diagnostic rate for syndromic pediatric patients, different factors such as capture and library kits, batch effects, DNA quality, and poor coverage regions may affect the efficiency of the WES-based CNV analysis [19]. CNVs larger than 1 mb can be well detected by WES CNV with high sensitivity and specificity [20]. In addition, there was no statistically significant correlation (*p* = 0.067) between the phenotype of severe lens abnormality and the genotype of *ARL14EP* haploinsufficiency, due to the limited sample size of WAGR patients. We will continue to collect phenotypic and genotypic data from WAGR patients and perform a methodologically sound functional study on the *ARL14EP* gene.

## 5. Conclusions

The present study expands the genotypic and phenotypic spectrum of WAGR syndrome. We identified six different de novo deletions located at 11p13 from six probands in our exon sequencing data of 269 probands with CCs. The associations between *ARL14EP* haploinsufficiency and rare phenotypes of severe lens abnormalities, lens automated absorption, and lens thinning were analyzed using a genotype–phenotype correlation analysis. Further functional studies are necessary to confirm the associations and unravel the pathogenetic mechanism.

## Figures and Tables

**Figure 1 genes-13-01431-f001:**
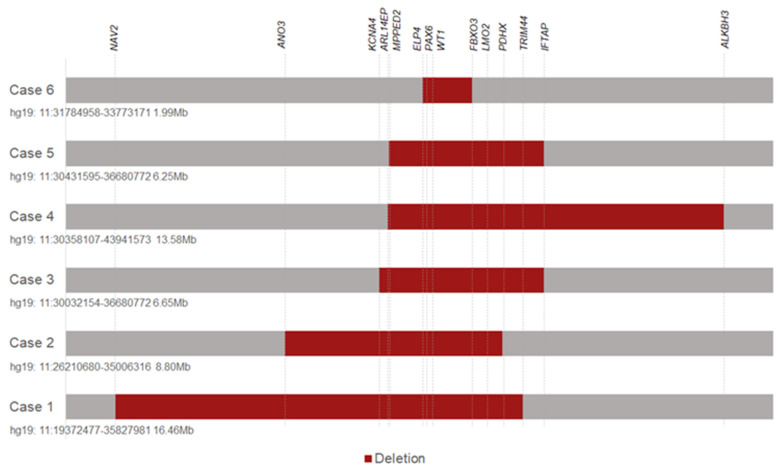
Chromosome 11p13 deletions from 6 patients with WAGR syndrome identified from whole exome sequencing data.

**Figure 2 genes-13-01431-f002:**
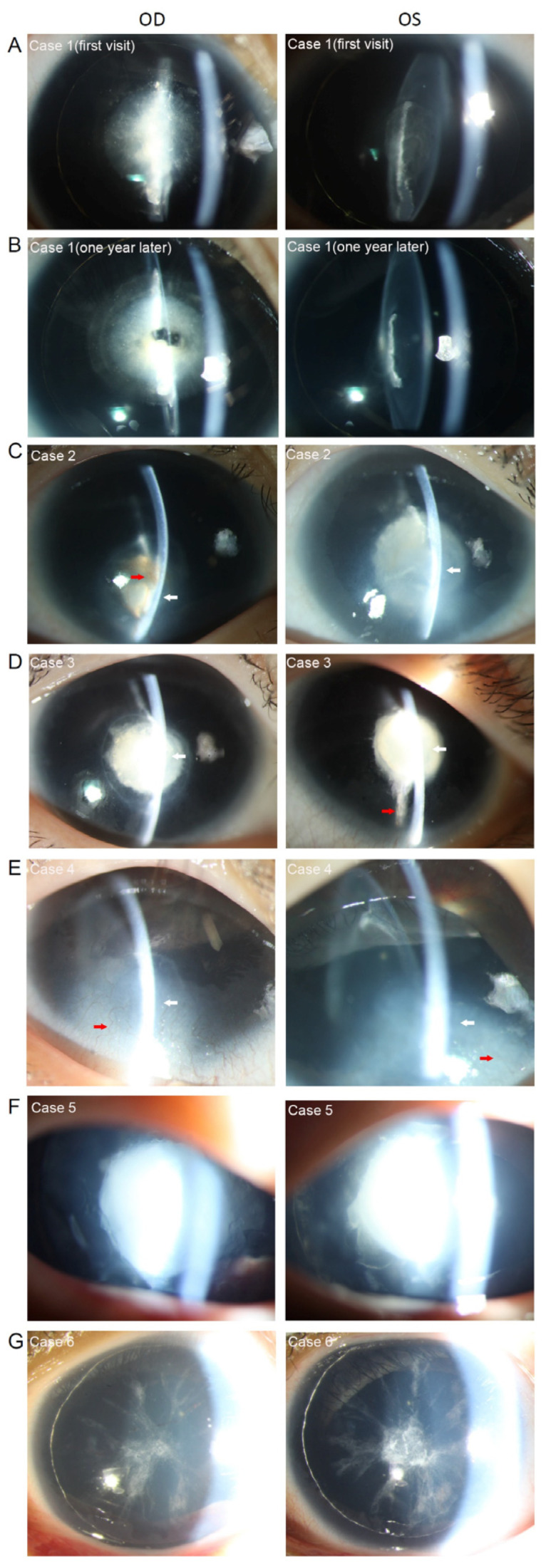
Anterior segment photographs of 6 patients with WAGR syndrome. (**A**). Anterior segment of case 1 at her first visit. (**B**). Anterior segment of case 1 one year later. (**C**). Anterior segment of case 2 showing keratolenticular adhesion, corneal neovascularization, aniridia, and cataract. (**D**). Anterior segment of case 3 showing keratolenticular adhesion, corneal neovascularization, aniridia, and the thin lens with cataract. (**E**). Anterior segment of case 4 showing keratolenticular adhesion, corneal neovascularization, aniridia, and the cataract. (**F**). Anterior segment of case 5 showing nuclear cataracts and aniridia. (**G**). Anterior segment of case 6 showing subcapsular cataract and aniridia. (White arrow: keratolenticular adhesion; red arrow: corneal neovascularization.).

**Figure 3 genes-13-01431-f003:**
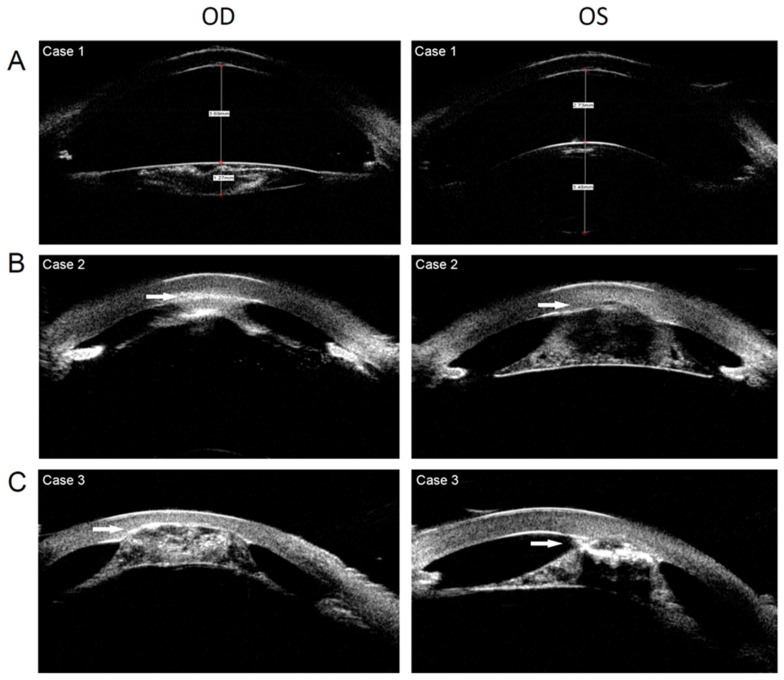
Ultrasound biomicroscopy of 3 patients with WAGR syndrome. (**A**). Anterior segment of case 1 showing the thin lens of the right eye and aniridia of both eyes. (**B**). Anterior segment of case 2 showing keratolenticular adhesion, aniridia, and thin lens. (**C**). Anterior segment of case 3 showing keratolenticular adhesion, aniridia, and thin lens. (White arrow: keratolenticular adhesion).

**Figure 4 genes-13-01431-f004:**
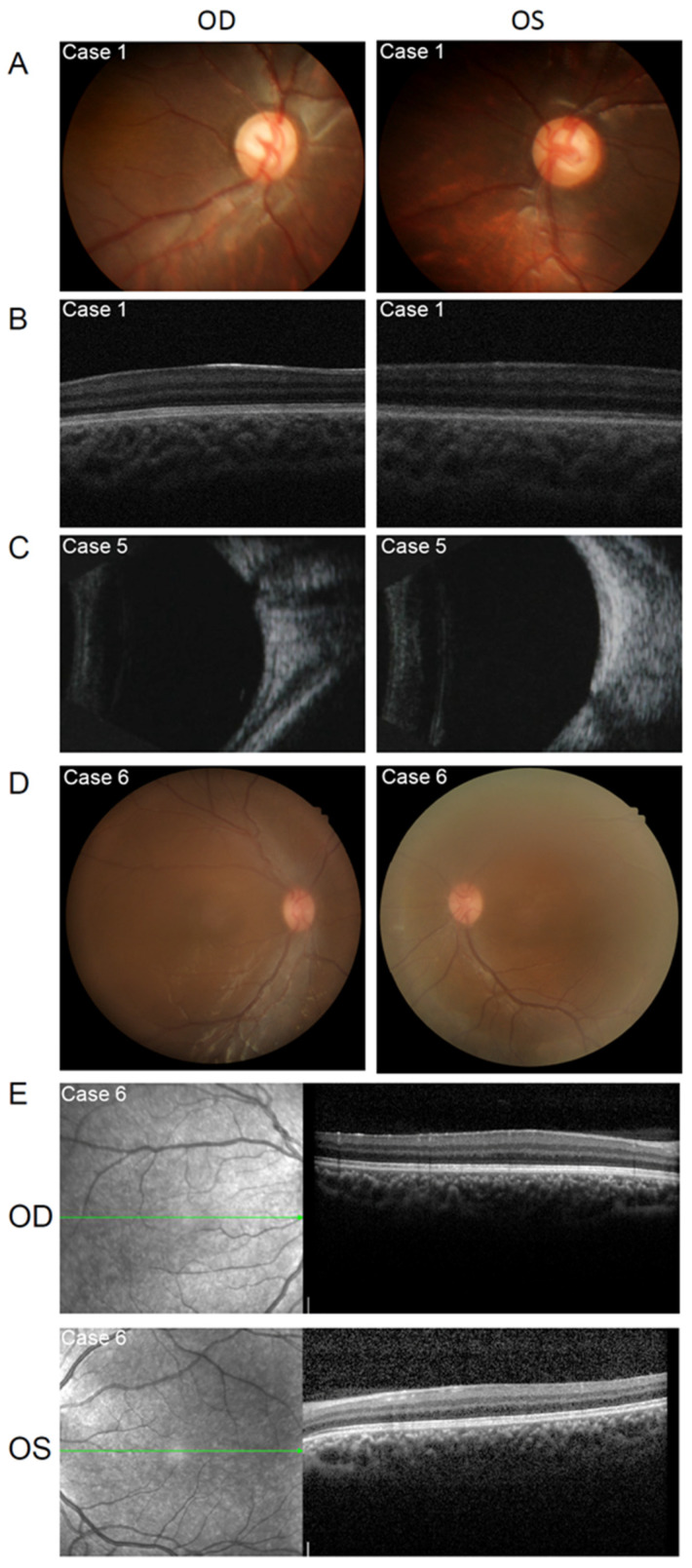
Ocular features of the patients with WAGR syndrome. (**A**). Fundus photograph of case 1 showing increased cup-to-disc ratio in both eyes. (**B**). OCT images of case 1 showing macular hypoplasia in both eyes. (**C**). B-scan images of case 5 showing increased cup-to-disc ratio in both eyes. (**D**). Fundus photograph of case 6 showing macular hypoplasia of both eyes. (**E**). OCT images of case 6 showing macular hypoplasia in both eyes.

**Table 1 genes-13-01431-t001:** Patient Profiles: Clinical and genetic information of the patients with WAGR syndrome.

ID	Ethnicity	Age *	Gender	Region	BCVA	IOP (mmHg)	AL (mm)	Cornea	Lens	Iris	Fundus	Wilms Tumor	Others	Deleted Genes
1	Asian	11M	Female	Chr11:19372477-35827981	NA	15.3/16.2	22.24/22.98	Clear	Nuclear opacity and peripheral bubbles of the right eye; posterior cortical cataract of the left eye	Aniridia	Large cup-to-disc ratio, macular hypoplasia	Bilateral kidney	Nystagmus, thin lens, glaucoma, developmental delay	*NAV2*, *DBX1*, *HTATIP2*, *PRMT3*, *SLC6A5*, *NELL1*, *ANO5*, *SLC17A6*, *FANCF*, *GAS2*, *LUZP2*, *ANO3*, *MUC15*, *SLC5A12*, *FIBIN*, *BBOX1*, *CCDC34*, *LGR4*, *LIN7C*, *BDNF-AS*, *BDNF*, *HSP90AA2P*, *KIF18A*, *MIR610*, *METTL15*, *KCNA4*, *FSHB*, *ARL14EP*, *MPPED2*, *DCDC1*, *DNAJC24*, *IMMP1L*, *ELP4*, *PAX6*, *RCN1*, *WT1*, *WT1-AS*, *EIF3M*, *CCDC73*, *PRRG4*, *QSER1*, *DEPDC7*, *LINC00294*, *CSTF3*, *HIPK3*, *KIAA1549L*, *CD59*, *FBXO3*, *LMO2*, *CAPRIN1*, *NAT10*, *CAT*, *ELF5*, *EHF*, *APIP*, *PDHX*, *CD44*, *SLC1A2*, *FJX1*, *TRIM44*
2	Asian	23M	Female	Chr11:26210680-35006316	NA	16/35	16.63/17.07	Adherent corneal leukoma	Thin lens with nuclear opacity	Aniridia	Macular hypoplasia	Right kidney	Nystagmus, the surface of the thin lens touches the corneal endothelium, corneal neovascularization, glaucoma, nanophthalmos, atrial septal defect, developmental delay	*ANO3*, *MUC15*, *SLC5A12*, *FIBIN*, *BBOX1*, *CCDC34*, *LGR4*, *LIN7C*, *BDNF-AS*, *BDNF*, *HSP90AA2P*, *KIF18A*, *MIR610*, *METTL15*, *KCNA4*, *FSHB*, *ARL14EP*, *MPPED2*, *DCDC1*, *DNAJC24*, *IMMP1L*, *ELP4*, *PAX6*, *RCN1*, *WT1*, *WT1-AS*, *EIF3M*, *CCDC73*, *PRRG4*, *QSER1*, *DEPDC7*, *LINC00294*, *CSTF3*, *HIPK3*, *KIAA1549L*, *CD59*, *FBXO3*, *LMO2*, *CAPRIN1*, *NAT10*, *CAT*, *ELF5*, *EHF*, *APIP*, *PDHX*
3	Asian	18M	Male	Chr11:30032154-36680772	NA	18/17	NA	Adherent corneal leukoma	Thin lens with nuclear opacity and peripheral bubbles	Aniridia	Macular hypoplasia	Left kidney	Nystagmus, the surface of the thin lens touches the corneal endothelium, corneal neovascularization, developmental delay	*KCNA4*, *FSHB*, *ARL14EP*, *MPPED2*, *DCDC1*, *DNAJC24*, *IMMP1L*, *ELP4*, *PAX6*, *RCN1*, *WT1*, *WT1-AS*, *EIF3M*, *CCDC73*, *PRRG4*, *QSER1*, *DEPDC7*, *LINC00294*, *CSTF3*, *HIPK3*, *KIAA1549L*, *CD59*, *FBXO3*, *LMO2*, *CAPRIN1*, *NAT10*, *CAT*, *ELF5*, *EHF*, *APIP*, *PDHX*, *CD44*, *SLC1A2*, *FJX1*, *TRIM44*, *LDLRAD3*, *COMMD9*, *PRR5L*, *TRAF6*, *RAG1*, *RAG2*, *IFTAP*
4	Asian	16M	Female	Chr11:30358107-43941573	NA	14.3/12	NA	Adherent corneal leukoma	Cortical opacity	Aniridia	Macular hypoplasia	N	Nystagmus, the surface of the thin lens touches the corneal endothelium, corneal neovascularization, developmental delay	*ARL14EP*, *MPPED2*, *DCDC1*, *DNAJC24*, *IMMP1L*, *ELP4*, *PAX6*, *RCN1*, *WT1*, *WT1-AS*, *EIF3M*, *CCDC73*, *PRRG4*, *QSER1*, *DEPDC7*, *LINC00294*, *CSTF3*, *HIPK3*, *KIAA1549L*, *CD59*, *FBXO3*, *LMO2*, *CAPRIN1*, *NAT10*, *CAT*, *ELF5*, *EHF*, *APIP*, *PDHX*, *CD44*, *SLC1A2*, *FJX1*, *TRIM44*, *LDLRAD3*, *COMMD9*, *PRR5L*, *TRAF6*, *RAG1*, *RAG2*, *IFTAP*, *LRRC4C*, *API5*, *TTC17*, *HSD17B12*, *ALKBH3*
5	Asian	3M	Female	Chr11:30431595-36680772	NA	12.2/14.6	15.83/13.98	Clear	Nuclear opacity	Aniridia	Large cup-to-disc ratio; macular hypoplasia	Bilateral kidney	Nystagmus, glaucoma onset after cataract extraction, nanophthalmos, developmental delay	*MPPED2*, *DCDC1*, *DNAJC24*, *IMMP1L*, *ELP4*, *PAX6*, *RCN1*, *WT1*, *WT1-AS*, *EIF3M*, *CCDC73*, *PRRG4*, *QSER1*, *DEPDC7*, *LINC00294*, *CSTF3*, *HIPK3*, *KIAA1549L*, *CD59*, *FBXO3*, *LMO2*, *CAPRIN1*, *NAT10*, *CAT*, *ELF5*, *EHF*, *APIP*, *PDHX*, *CD44*, *SLC1A2*, *FJX1*, *TRIM44*, *LDLRAD3*, *COMMD9*, *PRR5L*, *TRAF6*, *RAG1*, *RAG2*, *IFTAP*
6	Asian	10Y	Male	Chr11:31784958-33773171	0.1/0.15	13.3/14.2	15.55/15.42	Clear	Subcapsular opacity	Aniridia	Macular hypoplasia	N	Nystagmus, nanophthalmos, hypospadias, obesity	*ELP4*, *PAX6*, *RCN1*, *WT1*, *WT1-AS*, *EIF3M*, *CCDC73*, *PRRG4*, *QSER1*, *DEPDC7*, *LINC00294*, *CSTF3*, *HIPK3*, *KIAA1549L*, *CD59*, *FBXO3*

* Age: age at peripheral blood extraction; M: month; Y: year; BCVA: best corrected visual acuity; IOP: intraocular pressure; AL: axial length. NA: not available; N: none.

## Data Availability

The data are available from the corresponding authors upon reasonable request.

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
