# Peer review of "Unusual Presentation in WAGR Syndrome: Expanding the Phenotypic and Genotypic Spectrum of the Diseases"

_genes, 2022, doi:10.3390/genes13081431_

Round 1

Reviewer 1 Report

Qiwei et al., conducted a detailed clinical study on a patient cohort (269) affected with WAGR syndrome. Patients were enrolled in the study with basic inclusion criteria of congenital cataracts and completion of Whole exome sequencing at Childhood Cataract Program of the Chinese Ministry of Health. In the current study, authors have found various overlapping heterozygous deletions (involving different genes) at chromosomal position 11p13 in six unrelated patients. The rare clinical findings of the affected individuals include automated lens absorption and lens thinning along with typical WAGR syndromic features. Have highlighted the deletion of ARL14EP  gene in four patients and presumed it to further deteriorate the lens abnormalities in these patients.  Thus, expanding the genotypic spectrum of this severe disorder. This well-documented manuscript requires minor edits to get approve for publication.  

Introduction: Line 34, what is 6 with syndrome name? Could please fix reference issues, sometimes this is in brackets and sometimes not, I guess this is due to software used to generate reference library.

Material and Methods: Could please add reference in line 73, where author wrote as previously described. Could please write the manufactures of the equipment’s/machinery used for all clinical tests.   

Results: In the result section, authors have mentioned the deletion of ARL14EP in four probands, could please add those 4 case IDs there. 

Figures: Figure 1, could please add human assembly reference (top of the figure) and deletion region (chromosomal position and total deletion size, e.g., Mb or Kb) in front of case IDs. Could please add case/patient IDs on the figures (2 and 3 both) for the ease of readers. And add arrows/arrow heads to highlight phenotype e.g., keratolenticular adhesion or aniridia etc. Figure 2 legend, this is written photographer which need to be changed to photographs.  

Table: Could please add ethnicity of the affecteds in the clinical table.  Add borders in the table, this is hard to see which phenotype is for which case in “others” column. In last column, could please check it is others or orthers? Could please also add the deleted gene names in the table, for the readers convenience and phenotype comparison.  

Author Response

Point 1: Introduction: Line 34, what is 6 with syndrome name? Could please fix reference issues, sometimes this is in brackets and sometimes not, I guess this is due to software used to generate reference library.

Response 1: Thanks a lot for the suggestion. We have fixed the reference issue. (Line 34, highlighted in yellow)

Point 2: Material and Methods: Could please add reference in line 73, where author wrote as previously described. Could please write the manufactures of the equipment’s/machinery used for all clinical tests.

Response 2: We thank the reviewer for pointing this out. We have added the reference [11] in line 76 where we wrote as previously described, and written the manufactures of the equipment’s/machinery used for all clinical tests. (Line 64-69, line 76, highlighted in yellow)

Reference:

  1. Wang, X.; Wang, D.; Wang, Q.; Huang, W.; Dongye, M.; Zhang, X.; Lin, D.; Lin, Z.; Li, J.; Hu, W.; et al. Broadening the Mutation Spectrum in GJA8 and CHMP4B: Novel Missense Variants and the Associated Phenotypes in Six Chinese Han Congenital Cataracts Families. Front Med (Lausanne) 2021, 8, 713284, doi:10.3389/fmed.2021.713284.

Point 3: In the result section, authors have mentioned the deletion of ARL14EP in four probands, could please add those 4 case IDs there.

Response 3: As suggested, 4 case IDs were listed as followed: Case1, Case 2. Case 3 and Case 4. (Line 216, highlighted in yellow)

Point 4: Figures: Figure 1, could please add human assembly reference (top of the figure) and deletion region (chromosomal position and total deletion size, e.g., Mb or Kb) in front of case IDs. Could please add case/patient IDs on the figures (2 and 3 both) for the ease of readers. And add arrows/arrow heads to highlight phenotype e.g., keratolenticular adhesion or aniridia etc. Figure 2 legend, this is written photographer which need to be changed to photographs.

Response 4: We appreciate very much the constructive comments. We have added human assembly reference and deletion region in front of case IDs, case/patient IDs on the new figures (2 and 3 both) and arrows/arrow heads to highlight phenotype. The “photographer” in Figure 2 legend were changed to “photographs”. (Line 131-139, line 145, highlighted in yellow)

Point 5: Table: Could please add ethnicity of the affecteds in the clinical table. Add borders in the table, this is hard to see which phenotype is for which case in “others” column. In last column, could please check it is others or orthers? Could please also add the deleted gene names in the table, for the readers convenience and phenotype comparison.

Response 5: As suggested, we have revised the table as shown in new Table 1.

Reviewer 2 Report

The paper is interesting and brings new information. I suggest reviewing English and typing.

Minor corrections should be made:

Line 31 – replace Moreover for moreover

Line 35 – replace “syndrome6” for “syndrome[6]”

Line 41 – replace “studies1,9” for “studies[1,9]”

Line 46 – replace “families; the case” for “families, that revealed rare….”

Line 70 – replace “QI-Aamp DNA” for “QIAamp DNA”

The arrangement of images in figure 2 and 4 could be improved.

Author Response

Point 1: Line 31 – replace Moreover for moreover.

Response 1: Done. (Line 31, highlighted in yellow)

Point 2: Line 35 – replace “syndrome6” for “syndrome[6]”.

Response 2: Done. (Line 34, highlighted in yellow)

Point 3: Line 41 – replace “studies1,9” for “studies[1,9]”.

Response 3: Done. (Line 42, highlighted in yellow)

Point 4: Line 46 – replace “families; the case” for “families, that revealed rare….”.

Response 4: Done. (Line 46, highlighted in yellow)

Point 5: Line 70 – replace “QI-Aamp DNA” for “QIAamp DNA”.

Response 5: Done. (Line 74, highlighted in yellow)

Point 6: The arrangement of images in figure 2 and 4 could be improved.

Response 6: Thanks a lot for the suggestion. The arrangement of images in figure 2 and 4 have been improved as show in new figure 2 and figure 4.